# Human Induced Pluripotent Stem Cell Phenotyping and Preclinical Modeling of Familial Parkinson’s Disease

**DOI:** 10.3390/genes13111937

**Published:** 2022-10-25

**Authors:** Jeffrey Kim, Etienne W. Daadi, Thomas Oh, Elyas S. Daadi, Marcel M. Daadi

**Affiliations:** 1Southwest National Primate Research Center, Texas Biomedical Research Institute, San Antonio, TX 78227, USA; 2Cell Systems and Anatomy, San Antonio, TX 78229, USA; 3Department of Radiology, Long School of Medicine, University of Texas Health at San Antonio, San Antonio, TX 78229, USA

**Keywords:** Parkinson’s disease, genetic basis for pathophysiology, induced pluripotent stem cells, brain organoids, in vitro models of familial Parkinson’s disease, personalized medicine

## Abstract

Parkinson’s disease (PD) is primarily idiopathic and a highly heterogenous neurodegenerative disease with patients experiencing a wide array of motor and non-motor symptoms. A major challenge for understanding susceptibility to PD is to determine the genetic and environmental factors that influence the mechanisms underlying the variations in disease-associated traits. The pathological hallmark of PD is the degeneration of dopaminergic neurons in the substantia nigra pars compacta region of the brain and post-mortem Lewy pathology, which leads to the loss of projecting axons innervating the striatum and to impaired motor and cognitive functions. While the cause of PD is still largely unknown, genome-wide association studies provide evidence that numerous polymorphic variants in various genes contribute to sporadic PD, and 10 to 15% of all cases are linked to some form of hereditary mutations, either autosomal dominant or recessive. Among the most common mutations observed in PD patients are in the genes LRRK2, SNCA, GBA1, PINK1, PRKN, and PARK7/DJ-1. In this review, we cover these PD-related mutations, the use of induced pluripotent stem cells as a disease in a dish model, and genetic animal models to better understand the diversity in the pathogenesis and long-term outcomes seen in PD patients.

## 1. Parkinson’s Disease Pathology, Symptoms, Treatments

Parkinson’s disease (PD) is a chronic and progressive neurodegenerative disorder that arises from the loss of dopaminergic neurons in the substantia nigra pars compacta (SNpc) of the midbrain and other organ systems [1,2]. It was first characterized by Dr. James Parkinson in 1817 in his “An essay on the Shaking Palsy”, where he described patients with trembling extremities, hunched posture, and a shuffling gait [3]. It is the second most prevalent neurodegenerative disease that affects 1% of the population older than 60 years of age [4,5]. The number of PD patients has doubled from 1990 to 2015 and is projected to double again by 2040 [6]. At the onset of the disease, 30–70% of midbrain dopaminergic neurons are lost [7] yet the disease progresses slowly over years with a prodromal stage that may last 10–15 years before the presentation of motor deficits and proper diagnosis [8,9].

The loss of dopaminergic neurons in the SNpc severs the nigrostriatal pathway, which is one of four major dopamine pathways in the brain. When the connection between the SNpc and the dorsal striatum is lost, the ability to produce controlled movement becomes impaired. The symptoms of the disease include uncontrollable resting tremors, postural imbalance, cogwheel rigidity, bradykinesia, akinesia, and cognitive impairment [10,11,12,13,14,15,16]. Depression, dementia, and hallucinations are also observed [17,18,19,20,21,22,23,24,25]. Available treatments only provide symptomatic therapy. Levodopa or dopamine agonists are used to replace lost dopamine. However, long-term use of levodopa may lead to adverse side effects, such as dyskinesia [26]. Monoamine oxidase B (MAO-B) and catechol-O-methyl transferase (COMT) inhibitors are provided in conjunction with levodopa to inactivate dopamine metabolism and degradation [27]. When the patient no longer responds positively to dopamine replacement treatment, a neurosurgical procedure known as deep brain stimulation can be performed to alleviate symptoms [28,29,30,31,32]. This procedure utilizes an electrode that is surgically implanted into the ventral intermediate nucleus of the thalamus, globus pallidus, and subthalamic nucleus to reduce severe motor complications [33,34]. To date, there is no curative treatment available for PD.

For the vast majority of patients, the cause of PD remains unknown, and it is primarily an idiopathic disease. Nevertheless, there are many environmental risk factors that are known contributors to PD, including metals such as iron, copper, manganese, lead, and mercury, as well as toxins such as 1-methyl-4-phenyl-1,2,3,6-tetrahydropyridine (MPTP), rotenone, paraquat, dieldrin, hexachlorohexanes, and 2-4-dichlorophenoxyacetic acid [35]. The use of amphetamines and methamphetamines also increases risk [36]. However, age remains to be the strongest risk factor for PD [37].

At least 5% of cases are linked to specific genetic mutations [38]. Mutations in genes, such as leucine-rich repeat kinase 2 (LRRK2), α-synuclein (SNCA), glucosylceramidase β 1 (GBA1), phosphatase and tensing homolog-induced kinase 1 (PINK1), parkin (PRKN)and PARK7 (also referred to as DJ-1), are associated with increased PD risk [39]. Much of what we know about these mutations was discovered by using in vivo transgenic animal models and in vitro stem cell models [40,41,42,43]. The consequences of these mutations lead to aberrations in oxidative stress, mitochondrial dysfunction, perturbed protein quality control, protein aggregation, and altered kinase activity. Some mutations may also lead to the early onset of PD.

## 2. Leucine-Rich-Repeat Kinase 2 (LRRK2)

The LRRK2 protein has Roc-COR-kinase domains and exhibits profound kinase and GTPase activity [44,45,46]. It exists as an oligomeric structure with minimal kinase activity. Once bound to guanosine triphosphate (GTP), it dissociates to form an intermediate structure whereupon it autophosphorylates to form a homodimer kinase [47]. It has functions linked to transcription, translation, autophagy, mitochondrial function, cytoskeletal remodeling vesicular transport, dopamine homeostasis, and synaptogenesis [48,49,50,51,52,53]. It has also been shown in human brains to be constitutively expressed in neurons and glia [54]. LRRK2 interacts with 14-3-3 proteins via phosphorylated Ser910 and Ser935 [55]. The binding of 14-3-3 to LRRK2 is thought to be disrupted with PD-related mutations [56]. Knockout of wild-type LRRK2 causes impairment of protein degradation pathways, accumulation of α-synuclein (α-syn), and apoptotic cell death in aged mice, signifying LRRK2 is essential in those pathways [57]. Furthermore, increased LRRK2 kinase activity was observed in post-mortem brain tissue and immune cells of idiopathic PD patients, providing clinical significance of LRRK2 as a therapeutic target [58,59,60].

LRRK2 is known to be phosphorylated at these sites: Ser910, Ser935, Ser955, Ser973, Ser1292, and Thr826 [61]. Specifically, the Ser1292 residue is the site of autophosphorylation [62]. Autophosphorylation is significantly increased in disease-causing variants [63]. The kinase activity is also required for the cytotoxicity of LRRK2 mutants [64,65]. Autophosphorylation was observed to have decreased when LRRK2 was chemically inhibited [55]. Pharmacological inhibition was reported to protect against the toxic phenotype of hyper-kinase activity [66]. However, LRRK2 kinase inhibition was shown to have off-target effects in peripheral tissues in animal models [67,68,69].

LRRK2 was reported to phosphorylate cytoskeletal proteins, such as tau, microtubule affinity regulating kinase 1 (MARK1), tubulins, rho guanine nucleotide exchange factor 7 (ARHGEF7), and ezrin-radixin-moesin (ERM) [70,71,72,73,74,75,76]. LRRK2 also exhibits physical interaction with F-actin and microtubules [66,77]. Pathogenic LRRK2 is believed to alter cytoskeletal scaffolding leading to deficits in neurite outgrowth and axonal transport [78,79]. In fact, phosphorylation of ezrin, radixin, and moesin (*ERM*) family proteins by pathogenic LRRK2 promoted the cytoskeletal rearrangement [76]. Rab10 is another phosphorylation target of LRRK2, specifically at Thr73 [80]. LRRK2 variants augmented Rab10 phosphorylation which affected vesicular transport [81]. Increased Rab10 phosphorylation was observed in dopaminergic neurons in the SNpc of idiopathic PD patients, which makes this an excellent and indirect measure for the LRRK2 kinase activity [82]. LRRK2 was also shown to control the vesicle trafficking [83,84]. Snapin, EndophilinA, and Rab5a were identified as possible LRRK2 substrates involved in synaptic vesicle exo- and endocytosis [85,86,87,88]. Evidence suggests that LRRK2 could inhibit autophagy, causing accumulation of autophagic organelles, that is reversed through kinase inhibition [89,90,91]. Hyperphosphorylation also affected 4E-BP1, leading to excessive protein translation and neurodegeneration in Drosophila model [50]. However, Drosophila only possesses LRRK1 and not LRRK2 and such findings were not strongly translated to mammalian models [92]. Although, LRRK2 phosphorylates ribosomal protein S15, regulating protein synthesis in the G2019S LRRK2 transgenic Drosophila and human dopaminergic neurons [93].

## 3. Common LRRK2 Mutations That Lead to Parkinson’s Disease

The most prevalent genetic mutation has been identified in the LRRK2 gene [94,95]. This particular mutation, known as the G2019S, accounts for at least 1% of all PD cases and 5–15% of familial PD cases [96]. Clinically, symptoms of LRRK2 mutant carriers are indistinguishable from idiopathic PD [97]. LRRK2 variants either increase or decrease the risk of developing PD. The common variants in Asia associated with increased risk of PD are A419V, R1628P, and G2385R [98,99,100]. These missense variants increase the risk for disease by about two-fold [101]. Interestingly, the variants N551K, R1398H, and K1423K appear to reduce the risk of PD [102,103]. There are eight isoforms of LRRK2 that lead to autosomal dominant familial PD: R1441C/H/G/S, Y1699C, N1437H, I2020T, and G2019S [104,105,106,107]. Each of these isoforms is a single amino acid substitution. The R1441C/H/G/S, N1437H, and Y1699C variants are located in the GTPase domain and reduce the GTP hydrolysis [108,109,110]. The variants in the GTPase and kinase domains are linked to neurotoxicity [52,63]. The I2020T and G2019S mutations are found on the kinase domain and lead to increased LRRK2 kinase activity [63,64,65,107,111,112]. Although the effect on kinase activity is not as pronounced in I2020T as it is in the G2019S variant [113], it has been well established that the mutations within the kinase domain augment kinase function and lead to Parkinsonism.

## 4. The G2019S LRRK2 Mutation

The most prevalent of all LRRK2 mutations is the G2019S point mutation located in the kinase domain of the protein [95,114]. This mutation exhibits high prevalence in Ashkenazi Jewish, North African Berber, and Arab PD patients [115,116,117]. Interestingly, it is not as common among Asian PD patients [39,118,119,120,121,122,123]. It is an autosomal dominant point mutation that resides in the kinase domain. This in turn leads to the hyper-kinase activity of LRRK2 exhibited in enhanced autophosphorylation at S1292, up to four-fold increase [63,64,65,124]. However, the mutation does not alter the gene expression of LRRK2. The symptoms that arise in disease-manifesting carriers closely resemble those associated with idiopathic PD [125,126]. Kinase inhibition blocks neurotoxicity in vitro and in vivo [66]. Interestingly, there is no marked phenotypic difference between heterozygous and homozygous G2019S mutant carriers [127]. Dopaminergic neurons derived from induced pluripotent stem cells (iPSC) harboring the G2019S mutation demonstrate increased susceptibility to oxidative stress and early neuritic branching defects [128]. G2019S LRRK2 is also linked to an upregulation of the p53-p21 pathway, contributing to cellular senescence and accumulation of α-syn [129]. However, the full impact of G2019S mutation is still largely unknown [97].

## 5. G2019S LRRK2 Induced Pluripotent Stem Cell Models

In vitro disease modeling has been revolutionized with the use of iPSCs [130,131,132,133,134]. To generate iPSCs, somatic cells are reprogrammed with the transfection of pluripotent factors octamer-binding protein 3/4 (Oct3/4), SRY-box transcription factor 2 (Sox2), c-Myc, and Krüppel-like factor 4 (Klf4) [135]. The reprogramming of adult somatic cells allows for the creation of patient-specific iPSCs that harbor known pathological genetic mutations [136]. The reprogrammed cells express embryonic stem cell markers and are capable of differentiating into any cell type of the three germ layers. Patient iPSCs can be genetically modified to correct mutations with CRISPR/Cas9, transcription activator-like effector nucleases (TALENS), or zinc-finger nuclease [137,138,139,140,141,142]. iPSCs can then be differentiated into a cell type of choice and sequenced to identify differentially expressed genes. Patient-derived iPSCs are a powerful tool to study a disease in a dish, and indeed many groups have used neurons derived from PD patient iPSCs to advance our understanding of the cause of neuronal degeneration.

The first LRRK2 iPSCs were generated from the PD patient fibroblasts of a 63-year-old male homozygous carrier and a 42-year-old male heterozygous for G2019S LRRK2 [143]. Accumulation of α-syn and neurite retraction have been observed in LRRK2 G2019S cells [128,144,145]. Endogenous α-syn accumulation also occurred when exogenous α-syn in the form of recombinant human preformed fibrils was cultured in iPSC-derived neurons [146]. Neural stem cells derived from G2019S iPSCs exhibited increased susceptibility to proteasomal stress and passage-dependent deficiencies in nuclear-envelope organization, clonal expansion, and neuronal differentiation [143]. Specifically, neuronal differentiation was affected as dopaminergic neurons showed early neuritic branching defects [147]. Sensory neurons derived from iPSCs also showed shortened neurites as well as microtubule-rich axon aggregation and altered calcium dynamics [148]. Furthermore, G2019S iPSC-derived dopaminergic neurons also displayed lower baseline ER-Ca^2+^ levels with Ca^2+^ influx increased, and Ca^2+^ buffering capacity decreased after membrane depolarization [149].

Single-cell RNA-sequencing (scRNA-seq) of G2019S PD patient iPSC-derived neural stem cells revealed affected genes involved in mitochondrial function, DNA repair, protein degradation, oxidative stress, lysosome biogenesis, ubiquitin–proteasome system, endosome function, autophagy, and mitochondrial quality control [150]. It was also recently shown in another scRNA-seq experiment that neuroepithelial stem cells (NESC) derived from G2019S iPSCs exhibited mitochondrial defects [151]. They found that the G2019S NESCs exhibited fragmented mitochondria, impaired mitochondria function, and impaired autophagosomal-lysosomal pathway, and the cells were more prone to release reactive oxygen species (ROS). They later expanded on their scRNA-seq data to show that G2019S LRRK2 NESCs initiated early cell-cycle exit and earlier neural differentiation than wild type (WT), leading to increased cell apoptosis [152]. This was in part due to the downregulation of the core regulatory circuit transcription factor Nuclear Receptor Subfamily 2 Group F Member 1 (NR2F1) and altered distal super-enhancer activity. Importantly, using a single-cell longitudinal imaging platform, early studies demonstrated that Nrf2 expression in neurons directly mitigates toxicity induced by α-synuclein and mutations in LRRK2 and that this effect is time-dependent [153]. Furthermore, mitochondrial genome damage and mitochondrial transport-related PD pathogenesis were also present [154,155]. It is also believed that G2019S LRRK2 disrupts the interaction between the mitochondrial transport protein, Miro, to PINK1 and Parkin, arresting the movement of damaged mitochondria along the cytoskeleton and delaying mitophagy [156]. Dopaminergic neurons derived from G2019S iPSCs show this abnormal mitochondrial trafficking and distribution as well [157]. This group also revealed that despite high levels of sirtuin, there was a reduction of sirtuin deacetylase activity, nicotinamide adenine dinucleotide (NAD+), and protein lysine deacetylase activity, leading to bioenergy deficits.

In addition to mitochondrial deficits, hyper-kinase activity also disrupts the endocytosis of synaptic vesicles in iPSC-derived ventral midbrain neurons [158]. Transcriptomics and proteomics revealed that clathrin-mediated endocytosis was disrupted [159]. Specifically, endothelial cytokines I–III, dynamin-1, and various Rab proteins were significantly downregulated. Evidence suggests that the synaptic defects are due to LRRK2 phosphorylation of auxilin (DNAJC6) which causes differential clathrin binding and disrupts endocytosis [160]. This impaired endocytosis led to the accumulation of oxidized dopamine and caused reduced glucocerebrosidase activity and increased α-syn accumulation.

As mentioned before, LRRK2 G2019S mutant neurons were found to exhibit increased accumulation and release of α-syn [161]. Mutant iPSC-derived midbrain dopaminergic neurons also showed higher basal levels of LC3 II. This aberrant autophagy was reversed with the inhibition of mitochondrial fission with fission dynamin-related protein 1 (DRP1) peptide inhibitor p110 [162]. Additionally, LRRK2 phosphorylation of leucyl-tRNA synthetase (LRS) was shown to reduce leucine binding and impair autophagy, leading to protein misfolding and endoplasmic reticulum stress [163].

LRRK2 further plays a role in the innate and acquired immunity of the peripheral and central nervous system [164]. Interestingly, LRRK2 is highly expressed in macrophages and microglia [164]. Evidence suggests that LRRK2 controls the secretion of inflammatory mediators [164,165,166]. LRRK2 GTPase function is also implicated in the inflammatory response [167,168,169]. Indeed, LRRK2 mutant microglia and astrocytes exhibit increased inflammatory cytokine and chemokine production [170,171,172,173,174]. Knockout or pharmacological inhibition of LRRK2 alleviates this inflammatory response [175,176,177,178,179]. Furthermore, LRRK2 mutations that lead to α-syn accumulation also impair nuclear factor kappa B (NF-*κ*B) signaling in iPSC-derived neurons [180]. G2019S iPSC-derived astrocytes exhibited downregulation of matrix metalloproteinase 2 (mmp2) and transforming growth factor β1 (TGFβ1) [181]. Finally, LRRK2 may also play a role in hematopoiesis as G2019S LRRK2 iPSC-derived monocytes undergo accelerated production while CD14+CD16+ monocytes are reduced [182]. These mutant monocytes also exhibit migratory deficits.

## 6. G2019S LRRK2 Animal Models

LRRK2 animal models have provided insights into the regulation of protein translation, vesicle trafficking, neurite outgrowth, autophagy, and cytoskeletal dynamics [93,183,184]. Different models exhibit different clinical aberrations, such as degeneration of midbrain dopaminergic neurons, accumulation of α-syn, abnormal dopamine secretion, and behavioral deficits. LRRK2 overexpression in Drosophila led to the age-dependent loss of dopaminergic neurons and reduction of locomotor activity [185,186,187]. G2019S LRRK2 induced loss of photoreceptors and the impaired visual system was also observed [188]. The mutation also induced the mislocalization of tau in dendrites, causing degeneration [189]. Overexpression of G2019S LRRK2 in C. elegans also resulted in dopaminergic neuron degeneration, enhanced vulnerability to mitochondrial dysfunction, and inhibition of autophagy [190,191]. Transgenic mice expressing G2019S mutations exhibited motor deficits but with minimal evidence of neurodegeneration [192]. Only two groups have demonstrated the G2019S LRRK2 model of age-dependent loss of dopaminergic neurons in the SNpc [193,194]. Transgenic rats expressing the G2019S mutation exhibited oxidative stress in the striatum and SNpc with increased inducible nitric oxide synthase expression and abnormal morphology of SNpc dopaminergic neurons [195,196]. Unfortunately, genetic animal models display inconsistent phenotypes and do not fully replicate the human condition (neurodegeneration, Lewy body formation, and significant motor deficits), thus better models are needed [197].

## 7. Synuclein α (SNCA)

The SNCA gene encodes the presynaptic protein α-Synuclein (α-syn), which has been found to be localized in the nuclear membrane and synaptic vesicles [198]. The function of this protein is not well understood but current evidence suggests it participates in the axonal transport of synaptic vesicles by binding and transporting fatty acids [199,200,201]. It is also believed to participate in the differentiation and survival of dopaminergic neuron progenitor cells of mice and humans [202,203]. In fact, α-syn is expressed in the SNpc, especially in neurons containing neuromelanin, an insoluble granular pigment [204].

The misfolding of α-syn can lead to aberrant aggregation in the form of insoluble filaments and deposits in nerve cells [205]. These α-syn aggregates are also a major component of Lewy bodies, cellular inclusions in the neuronal cytoplasm that may lead to PD pathogenesis [206]. Lewy bodies impair neuronal communication and may even spread to healthy neurons [207,208]. They are also known to increase oxidative stress, disrupt axonal transport for neurotransmitter vesicles, and contribute to transcriptional dysregulation, protein sequestration, mitochondrial and synaptic dysfunction, and inhibition of the ubiquitin–proteasome system [207,208,209,210,211,212,213]. Accumulation of α-syn also affects the lysosomal clearance of protein aggregates, thus resulting in a vicious cycle perpetuating the toxic effects of α-syn aggregates [214]. It has been shown that the majority of α-syn in Lewy bodies in postmortem PD brain tissue appears as phosphorylated S129 α-syn [215,216]. Evidence suggests that S129 phosphorylation promotes α-syn aggregation and neurotoxicity [217]. Aggregated forms of α-syn are in fact more prone to S129 phosphorylation and accumulation during disease progression [218,219,220]. Polo-like kinases, casein kinases, and G protein-coupled receptor kinases have been shown to modulate the phosphorylation of α-syn at S129 [220,221,222,223,224,225]. Interestingly, numerous reports suggest that LRRK2 may be involved in phosphorylating S129 and α-syn aggregation [226,227]. Ultimately, it is still uncertain whether S129 α-syn is indeed neurotoxic [228].

Braak et al. hypothesized that neurodegeneration occurs in a predetermined sequence caused by an unknown pathogen in the gut or nasal cavity initiating sporadic PD [229,230]. This is associated with a specific α-syn spreading pattern, which may be why PD patients exhibit gastrointestinal and olfactory problems [231,232,233]. Indeed, Lewy body pathology has been confirmed in neurons of the olfactory tract and enteric nervous system [234,235,236,237]. Therefore, it is possible that α-syn propagates in a prion-like fashion [238]. In fact, the cell-to-cell transmission of α-syn has been observed after transgenic α-syn overexpression or exposure to preformed fibrils of α-syn and homogenates from postmortem PD patients [239,240,241,242,243,244]. Therefore, the pathological aggregation of misfolded α-syn is hypothesized to be critical in PD pathogenesis.

An increased risk of developing PD in humans can result due to an overexpression of the SNCA gene because of locus triplication [245]. Alternatively, SNCA missense mutations have also been shown to increase PD risk. Mutations in the SNCA gene were first identified as causing autosomal dominant PD in a large Italian family known as the Contursi kindred [246]. These patients carried the A53T point mutation and exhibited hallmark characteristics of PD including Lewy body pathology and positive response to L-dopa treatment. However, early onset and rapid disease progression were also observed. Other missense mutations include A30P [247,248,249], E46K [250], A53E [251,252], A53V [253], G51D [254,255], H50Q [256], and A18T and A29S [257]. Copy number variations have also been reported in PD patients [245]. Carriers of the A30P mutation were also associated with early onset, but a milder disease progression was observed compared to those with the A53T mutation [247]. The A53T and A30P mutations increase the likelihood of α-syn protein oligomerization instead of fibrillation [258], which is believed to accelerate α-syn aggregation [259]. Patients with A53T and E46K developed dementia consistent with Lewy body dementia [250,260] as the E46K mutation also promotes α-syn aggregation similar to A53T [261]. Interestingly, patients with the E46K mutation have also experienced visual hallucinations [250]. The A18T and A29S missense mutations were found in patients with sporadic PD [257].

## 8. SNCA Induced Pluripotent Stem Cell Models

Dopaminergic neurons derived from iPSCs of a PD patient with SNCA triplication have been studied. The level of α-syn protein in these dopaminergic neurons was twice the amount compared to those derived from normal iPSCs [262]. These neurons exhibited changes in growth, viability, cellular energy metabolism, and stress resistance when challenged with starvation or toxins [263], and increased oxidative stress [264]. SNCA triplication also led to a reduced capacity for iPSCs to differentiate into neurons, decreased neurite outgrowth, and lower neuronal activity compared to normal neurons [265]. The mRNA levels of nuclear receptor-related 1 protein (NURR1), G-protein-regulated inward-rectifier potassium channel 2 (GIRK-2), and tyrosine hydroxylase (TH) were also significantly reduced. Furthermore, lysosomal dysfunction has also been also induced by the α-syn accumulation [266]. Aggregates appeared to interact with ATP synthase and lead to premature mitochondrial permeability transition pore opening, making neurons more vulnerable to cell death [267]. Interestingly, SNCA triplication also affects non-neuronal cells, such as microglia. Microglia derived from SNCA triplication iPSCs exhibited impaired phagocytosis compared to isogenic controls [268].

An isogenic gene-corrected iPSC line for A53T was generated with zinc-finger nuclease-mediated genomic editing [140]. Gene correction reversed nitrosative stress and endoplasmic reticulum stress in iPSC-derived neurons [269]. A53T mutation increased apoptotic cell death in iPSC-derived midbrain dopaminergic neurons by increasing S-nitrosylation of MEF2C, affecting the transcriptional regulation of PGC1a, a master regulator of mitochondrial biogenesis [270]. A53T iPSC-derived neurons also displayed irregular protein aggregation, compromised neuritic outgrowth, contorted or fragmented axons with varicosities containing α-syn and Tau, and disrupted synaptic connectivity [271]. Interestingly, A53T midbrain dopaminergic neurons contained higher concentrations of α-syn monomers relative to tetramers when compared to isogenic controls, ultimately decreasing solubility α-syn [272]. Oligomeric α-syn has emerged as the key mediator for α-syn accumulation [273]. In fact, A53T and SNCA triplication iPSC-derived dopaminergic neurons exhibited increased α-syn oligomerization in a proximity ligation assay [274]. Increased sensitivity to mitochondrial toxins and nitrosative stress-induced neuronal loss were also observed in A53T dopaminergic neurons [275]. Furthermore, transcriptomic analysis of A53T and SNCA triplication iPSC-derived dopaminergic neurons revealed perturbations in the expression of genes likened to mitochondrial function, which was consistent with a reduction in mitochondrial respiration, impaired mitochondrial membrane potential, aberrant mitochondrial morphology, and decreased levels of phosphorylated DPR1 Ser616 [274]. They also observed increased endoplasmic reticulum stress and impaired cholesterol and lipid homeostasis. Single-cell transcriptomic analysis of A53T iPSC-derived dopaminergic neurons compared to an isogenic gene-corrected counterpart revealed perturbations in glycolysis, cholesterol metabolism, synaptic signaling, and ubiquitin–proteasomal degradation [276]. Apart from neuronal lineage cells, O4+ oligodendrocyte linage cells derived from A53T iPSCs also exhibited impaired maturation compared to controls [277].

## 9. SNCA Animal Models

Transgenic models with SNCA mutations have typically failed to display clear dopaminergic neurodegeneration or parkinsonian motor deficits [278]. However, there has been apparent α-syn aggregation and altered neuronal functions in these models. For example, transgenic mice expressing truncated α-syn show a reduced number of nigro-striatal neurons due to cell loss during early development [279]. However, the clinical relevance of these transgenic models remains questionable as the mutated α-syn protein is affecting early developmental stages rather than the later onset of neurodegeneration as seen in human patients [280].

SNCA overexpression models in rodents have been shown to affect the development and maintenance of dopaminergic neurons [281]. Overexpression has led to the formation of α-syn aggregates in the brain causing motor and olfactory deficits but not dopaminergic neurodegeneration [282]. E46K SNCA rats also display α-syn aggregation, altered metabolism of striatal dopamine, and increased oxidative stress but again not dopaminergic neurodegeneration [283].

Viral vectors such as adeno-associated virus (AAV) and lentivirus (LV) have also been used for transfecting rodents with SNCA. The AAV6 serotype generated an 80% loss of dopaminergic neurons and profound motor deficits [284]. This model also showed progressive neurodegeneration over a 2-to-4-month period. Using an LV vector led to α-syn aggregation but with no apparent loss of dopaminergic neurons nor behavioral changes [285]. Alternatively, AAV2/7-α-syn transduction in mouse SNpc produced dose-dependent dopaminergic neurodegeneration and motor deficits [286]. AAV2/2 was used to deliver WT and A53T α-syn into marmosets, leading to 30–60% nigral dopaminergic neurodegeneration and subsequent striatal dopamine depletion with mild motor deficits [287]. AAV1/2-A53T α-syn in macaques displayed 30% nigrostriatal dopaminergic neurodegeneration, 50% dopamine depletion, and 40% DAT reduction [288].

## 10. Glucocerebrosidase 1 (GBA1)

GBA1 encodes a lysosomal protein β-glucocerebrosidase (GCase). Mutations in this gene result in the accumulation of glycolipid substrates in lysosomes which disrupts lysosomal function and can lead to an autosomal recessive lysosomal storage disorder known as Gaucher’s disease (GD) [289]. Although a small minority of GBA1 mutation carriers develop PD [290,291], mutations in GBA1 increase the risk of developing PD [292,293]. In fact, GBA1 mutations are common genetic risk factors for PD, where 7–10% of patients with PD are carriers of a GBA1 mutation [291,294]. The L444P point mutation has been identified as being associated with PD and GD patients [291,295]. Furthermore, the L444P mutation appears to have a higher risk of developing PD compared to other GBA1 point mutations [296,297,298]. Interestingly, there is no difference in risk between patients with either homozygous or heterozygous mutations, although homozygous carriers tend to have PD onset 6 to 11 years earlier than heterozygous counterparts [290,294,298,299]. The clinical features of GBA1-associated PD are similar to those associated with idiopathic PD, including olfactory deficits and sleep disturbance but with earlier onset and accelerated autonomic, cognitive, and motor decline [298]. These patients also present with a more pronounced loss of nigrostriatal dopaminergic neurons and greater Lewy body pathology compared to those with idiopathic PD. GCase deficiency and lysosomal dysfunction as a result of GBA1 mutations are thought to be important pathogenic mechanisms for PD [300]. GBA1 mutations appear to impair α-syn degradation in the lysosome due to perturbed GCase activity [301,302]. Interestingly, GCase activity is also reduced in postmortem brain tissue of PD patients without GBA1 mutations [303,304]. It is possible that the GlcCer substrate accumulation may promote the pathogenic conversion of α-syn into its insoluble form [305,306]. Indeed, the GlcCer substrate also stabilizes a-syn oligomeric intermediates and induces rapid polymerization of fibrils [302].

## 11. GBA1 Induced Pluripotent Stem Cell Models

Interestingly, WT GCase activity is reduced in brain tissue and iPSC-derived neurons of idiopathic PD patients and other genetic forms of PD without GBA1 mutation [160,303,304,307,308]. It was noted that mitochondrial oxidative stress leads to the accumulation of oxidized dopamine, resulting in reduced GCase activity, lysosomal dysfunction, and α-syn accumulation [309]. The iPSCs were generated from patients with GD and PD harboring GBA1 mutations and differentiated into midbrain DA neurons. The iPSC-derived neurons exhibited reduced GCase activity and protein levels, increased glucosylceramide and α-syn levels as well as autophagic and lysosomal defects [310]. The mutant neurons also exhibited dysregulation of calcium homeostasis and increased vulnerability to stress responses involving the elevation of cystolic calcium. Gene correction of the mutation rescues the observed pathological phenotypes [310]. N370S iPSC-derived dopaminergic neurons exhibited disruption of the autophagy pathway and ER stress leading to elevated extracellular a-syn [311]. GBA-PD patient-derived dopaminergic neurons with heterozygous N370S and L444P mutations display stress responses, sphingolipid accumulation, mitochondrial dysfunction, increased mitochondrial ROS, and changes in NAD+ metabolism, ameliorated with NAD+ precursor nicotinamide riboside [312]. Heterozygous-null GBA1 iPSC-derived cortical neurons and astrocytes exhibit reduced lysosome number, increased lysosomal pH, reduced lysosomal cathepsin protease activity, and increased accumulation of soluble and insoluble α-syn without changes in α-syn mRNA levels [313,314].

GCase chaperones were able to recover GCase activity and reduce α-syn levels in iPSC-derived DA neurons and mouse models [308]. GBA1 iPSC-derived neurons exhibited prolonged mitochondria–lysosome contacts due to defective modulation of the untethering protein TBC1D15, which mediates Rab7 GTP hydrolysis for contact untethering, ultimately leading to disrupted mitochondrial distribution and function [315]. This defect was rescued with a GCase modulator, indicating deficits were due to a lack of GCase activity. Another GCase chaperone S-181 was tested on 84GG GBA1 patient-derived dopaminergic neurons partially restored lysosomal function and lowered accumulation of oxidized dopamine, glucosylceramide, and α-syn [307].

Interestingly, G2019S LRRK2 iPSC-derived dopaminergic neurons also exhibited reduced GCase activity [316]. In fact, pharmacological inhibition of LRRK2 kinase activity increased GCase activity in both iPSC-derived dopaminergic neurons carrying LRRK2 and GBA1 mutations. The increase in GCase activity was sufficient to partially rescue the accumulation of oxidized dopamine and α-syn. Heterozygous-null GBA-1 iPSC-derived cortical neurons did not exhibit any differences in WT LRRK2 kinase activity [314]. However, LRRK2 inhibition rescued lysosomal number and Cathepsin L activity, and partial lysosome re-acidification. Interestingly, they did not see any change in GCase activity with LRRK2 inhibition, contradicting previous findings [316]. The possible interplay between LRRK2 and GBA1 is fascinating and needs to be further explored.

## 12. GBA1 Animal Models

There are over 200 known GD-associated mutations; therefore, it is difficult to determine which mutations are particularly responsible for PD susceptibility [295]. Homozygous L444P mice generated partial gene duplication and either died soon after birth due to compromised epidermal permeability barrier caused by defective glucosylceramide metabolism or exhibited systemic inflammation [317,318]. L444P conditional knock-in mice lived longer and exhibited increased striatal α-syn levels and astrogliosis at 1 year of age, although motor performance was not assessed [319]. Heterozygous L444P mice demonstrated impaired neuronal autophagy and mitophagy as well as mitochondrial dysfunction [320,321]. Heterozygous L444P mice also demonstrated impaired α-syn degradation and increased α-syn levels but interestingly did not form α-syn aggregates [320,321,322]. Furthermore, heterozygous L444P mice did not exhibit nigrostriatal neurodegeneration, neuroinflammation, or impairments in olfaction, coordination, and cognition [321,322,323]. The D409H mutation is a rare but severe mutation found in GBA1 PD patients [324]. α-Syn aggregation was observed in the cerebellum and brainstem of homozygous D409H mice [325]. However, heterozygous and homozygous D409H mice did not exhibit nigrostriatal neurodegeneration, neuroinflammation, or motor deficits [326]. Treatment with a GCase chaperone resulted in the activation of WT GCase and reduction of GCase lipid substrates and α-syn in brain tissue [307]. Homozygous V394L mice have 27% of WT GCase activity but do not exhibit α-syn aggregation [325]. Homozygous R643C mice exhibited increased nigrostriatal α-syn and neuroinflammation [319]. Homozygous N370S mutation has been neonatal lethal despite mild phenotype in patients [327]. Furthermore, several GBA1-PD models combine GBA1 mutation with overexpression of α-syn mutations to induce α-syn aggregation and pronounced PD symptoms despite the fact that these two mutations are not reported in PD patients [323,326,328].

## 13. PTEN-Induced Kinase 1 (PINK1)

Phosphatase and tensin homolog (PTEN)-induced putative kinase 1 (PINK1) is a mitochondria-targeted Ser/Thr protein kinase. In normal physiological conditions, PINK1 is imported into the mitochondria via the translocase of the outer membrane and translocase of the inner membrane [329]. PINK1 is cleaved in the transmembrane segment by the mitochondrial intramembrane protease PARL, where it is then retranslocated to the cytosol for proteasomal degradation [330]. Once the mitochondrial inner membrane becomes depolarized, PINK1 mitochondrial transport is arrested and phosphorylates Parkin on the ubiquitin-like domain and activates the E3 ligase activity of Parkin [331,332]. Parkin ubiquitinates mitochondrial outer membrane proteins and induces autophagic clearance of depolarized mitochondria [333,334,335]. PINK1 is localized on the mitochondria and may exhibit a protective effect, but as a result of a mutation that protection is lost, resulting in increased susceptibility to cellular stress [336]. It is also known to phosphorylate TNF receptor-associated protein 1 (TRAP1), 5-hydroxytryptamine receptor 2a (5-HT2A), and Parkin [337,338,339]. PINK1 mutations are the second most common autosomal-recessive form of early onset PD [329,340,341]. Furthermore, Pink1 mutation carriers may develop psychiatric comorbidity alongside gait disturbances [342,343].

## 14. PINK1 Induced Pluripotent Stem Cell Models

PINK1 kinase activity was significantly reduced in G411S PINK1 mutant iPSC-derived neurons [344]. In iPSC-derived neurons, PINK1 mutations reduced complex I activity, which lead to a reduction in mitochondrial membrane potential [345]. PINK1 PD patient iPSC-derived neurons treated with valinomycin, which triggers rapid loss of mitochondrial membrane potential, impaired recruitment of overexpressed Parkin to mitochondria [346,347]. Interestingly, loss of PINK1 abolished the degradation of mitochondrial protein only in fibroblasts, but not in isogenic iPSC-derived neurons, which do not exhibit significant mitophagy even with parkin overexpression and valinomycin treatment [346]. Phosphorylation of Ser250 in NdufA10 regulates the activity of ubiquinone reductase in mitochondrial complex I. A phosphomimetic mutant of NdufA10 reversed the deficiencies in complex I activity and ATP synthesis in PINK1 iPSC-derived neurons [345]. Furthermore, direct supplementation of cardiolipin, a mitochondrial inner membrane-specific lipid, to isolated mitochondria rescues PINK1-induced complex I defects [348]. Additionally, antioxidant treatment with coenzyme Q10 can rescue the cellular vulnerability associated with mitochondrial dysfunction in iPSC-derived neurons from PINK1 PD patients [349]. Apart from abnormal mitochondrial morphology, PINK1 PD patient iPSC-derived midbrain dopaminergic neurons exhibited α-syn accumulation and increased cytosolic dopamine levels [350]. Interestingly, increased expression of LRRK2 mRNA and protein was observed in PINK1 iPSC-derived neurons [351]. Moreover, transient overexpression of WT PINK1 can downregulate LRRK2 expression. The implication of this study suggests a convergent pathway between these two genes in PD pathogenesis.

## 15. PINK1 Animal Models

PINK1 knockout mice appear to have an age-dependent and moderate reduction in striatal dopamine levels accompanied by low locomotor activity due to deficient mitochondrial respiration and increased sensitivity to oxidative stress [352,353]. This model does not exhibit significant neurodegeneration, Lewy body formation, or loss of dopamine. Another PINK1 knockout mouse model has been reported to show olfactory and gait disturbances, similar to prodromal symptoms of human PD patients [354]. Pink1 deficient mice also have impaired dopamine release [355] and exhibited impaired complex I function, mitochondrial depolarization reduced ATP synthesis, and increased sensitivity to apoptotic stress [356]. As such, PINK1 knockout mice may be useful as a model for prodromal PD [357,358]. On the other hand, PINK1 null mice with an exon 4–5 deletion showed progressive loss of striatal dopamine, but nigrostriatal neurodegeneration was not observed [359].

The phenotype for rats differs from mice, with a closer resemblance to PD pathology. PINK1 knockout rats exhibit age-dependent loss of nigral dopaminergic neurons beginning at age 6–8 months [360,361]. Motor deficits including reduced rearing frequency and distance traveled in an open field, reduced hind limb grip strength, and increased foot slips and traversal time on a tapered balance beam were apparent [360,362]. These rats also exhibited mitochondrial respiration deficits and α-syn aggregation, although different from Lewy body pathology [361,362].

## 16. Parkin (PRKN)

Parkin is a RING domain-containing E3 ubiquitin ligase important for mitochondria quality control through mitophagy [363]. Autosomal recessive juvenile parkinsonism was first identified in Japanese patients with early onset PD [364]. Currently, there are more than 100 known mutations identified in Parkin [365,366,367,368]. Parkin mutations account for 50% of familial and 15% of sporadic cases of European PD patients with onset before 45 years of age [369,370]. Parkin mutations are also the most common form of juvenile PD or early onset PD [365,366,369,371,372,373]. Carriers of Parkin mutations exhibit earlier and more symmetrical onset, slower disease progression, and greater response to L-dopa, but seem to develop dyskinesia earlier as well [369,374,375]. Furthermore, cognitive impairment is rare, and dementia or depression are not present in Parkin mutation patients [369,376,377]. Most PD patients with Parkin mutations do not develop Lewy body pathology [378]. Furthermore, it appears that missense mutations are equally as detrimental to truncation and deletion mutations [379].

## 17. PRKN-Induced Pluripotent Stem Cell Models

Parkin patient iPSC-derived neurons present abnormal or enlarged mitochondria accompanied by increased oxidative stress and enhanced activity of nuclear factor erythroid 2-related factor 2 (Nrf2) pathway [380]. Proteomics analysis of Parkin knockout iPSC-derived neurons revealed disturbances in oxidative stress defense, mitochondrial respiration and morphology, cell cycle control, and cell viability [381]. Structural and functional analysis verified an increase in mitochondrial area and the presence of elongated mitochondria as well as impaired glycolysis and lactate-supported respiration. This abnormal mitochondria phenotype, such as elongated shape and larger volume, has been observed by other groups as well [350,380,382]. However, two different studies using qPCR showed that there was no significant difference in mitochondria DNA copy number [380,383]. Additionally, it seems Parkin mutations in iPSC-derived neurons may alter mitochondrial morphology in a portion of cells, particularly TH+ dopaminergic neurons [382]. In fact, Parkin patient iPSC-derived dopaminergic neurons exhibited smaller and less functional mitochondria than those in non-dopaminergic neurons [384] and exhibited increased cytotoxicity with known PD environmental risk factors, such as exposure to heavy metals such as copper and cadmium [385]. They also showed a significant increase in mitochondrial fragmentation, initial ROS generation, and loss of mitochondrial membrane potential following copper exposure. Transfected Parkin cell lines have shown that Parkin is recruited to the mitochondria to ubiquitinate a variety of substrates for the induction of mitophagy [363,386]. However, the recruitment of endogenous Parkin to mitochondria has not been robustly seen in cell lines [387,388], mice [389,390], or iPSC-derived human neurons [346,383].

The precision of dopaminergic transmission is significantly disrupted by increased spontaneous dopamine release and decreased reuptake [383]. Dopamine induces oscillatory neuronal activities in Parkin iPSC-derived neurons but not in normal iPSC-derived neurons [391]. Interestingly, this phenotype mirrors the widespread rhythmic bursting of neuronal activities in the basal ganglia of PD patients [392]. Parkin iPSC-derived midbrain dopaminergic neurons exhibited reduced length and complexity of neuronal processes understood to be caused by a marked decrease in microtubule stability, as the phenotype was rescued by taxol, a microtubule-stabilizing drug, or the overexpression of Parkin [393]. This decreased neurite length and complexity was also mimicked in normal neurons treated with colchicine, a microtubule-depolymerizing agent. Reduced microtubule stability in iPSC-derived neurons was observed by other groups as well [394].

In iPSC-derived neurons from two Parkin patients, increased accumulation of α-syn was observed in the patient with Lewy body pathology, but not in the other patient who did not have Lewy body pathology [380]. There was no significant difference in α-syn protein levels in iPSC-derived midbrain dopaminergic neurons from two Parkin PD patients and two normal subjects [383]. Although, an increased level of α-syn protein was observed in two other studies using iPSC-derived neurons from PD patients with Parkin mutations [350,382]. The inconsistency of α-syn accumulation in Parkin PD patient-derived cells may suggest that α-syn accumulation is independent of Parkin as Parkin PD patients do not necessarily develop Lewy body pathology.

## 18. PRKN Animal Models

Transgenic rodent models with Parkin overexpression exhibited protection from 6-OHDA or α-syn overexpression in the SNpc [395,396,397,398]. Parkin knockout mice have nigrostriatal mitochondrial respiration deficits and increased markers for oxidative stress but did not exhibit any significant nigrostriatal dopaminergic neurodegeneration nor PD-like locomotor deficits [360,399,400,401,402,403,404,405]. Although there were instances of Parkin knockout mice exhibiting slight impairment of dopamine release [406,407], they did not show significant behavioral deficits or age-dependent nigral dopaminergic neurodegeneration [360]. They also lacked Lewy body pathology similarly to human parkin mutant carriers that rarely showed LB pathology [408]. Successful transgenic rodent models that recapitulated dopaminergic neurodegeneration were eventually developed. Indeed, Parkin-Q311X-DAT-BAC mice expressing a C-terminal truncated human mutant Parkin in dopaminergic neurons exhibited late onset and progressive hypokinetic motor deficits, age-dependent nigrostriatal dopaminergic neurodegeneration, α-syn accumulation, and a significant reduction in striatal dopamine neuron terminals and dopamine levels [409]. Overexpression of both T240R Parkin and human WT Parkin in rats with AAV2/8 induced progressive and dose-dependent dopaminergic neurodegeneration [410].

## 19. Parkinsonism Associated Deglycase (DJ-1 Also Known as PARK7)

The DJ-1 protein is 189 amino acid residues with three cysteines and normally forms a homodimer that exhibits antioxidant activities [411]. DJ-1 is highly expressed in astrocytes in the frontal cortex and SNpc of idiopathic PD brains and is not an essential component of Lewy bodies [412,413]. It is also involved in the regulation of apoptosis and pro-survival signaling, autophagy, inflammatory responses, and protection against oxidative stress [414,415]. In fact, DJ-1 overexpression protects against oxidative stress while DJ-1 knockout increases oxidative stress-induced cell death [416,417,418,419,420,421]. It seems that the cysteine residue at position 106 is required for DJ-1 mediated protection from oxidative stress [411,422,423,424]. DJ-1 levels increase in response to oxidative stress caused by dopamine to suppress ROS accumulation [425]. DJ-1 also functions as a redox-sensitive chaperone that is activated in an oxidative cytoplasmic environment and can inhibit the generation of α-syn aggregates [426,427]. It can directly interact with α-syn monomers and oligomers, where mutant DJ-1 exhibits α-syn dimerization [428]. Furthermore, DJ-1 deficiency decreases LAMP2A expression, a receptor required for CMA-mediated α-syn degradation, thus leading to α-syn accumulation [429]. It can also act as a glyoxalase III to detoxify reactive dicarbonyl species, such as glyoxal and methylglyoxal into glycolic or lactic acid in the absence of glutathione [430].

A deletion mutation in a Dutch family and a homozygous point mutation L166P in an Italian family were identified to cause parkinsonism [431]. DJ-1 mutations only account for less than 1% of all early onset PD cases [432]. The median age of onset for DJ-1 PD is 27 years [433]. PD patients carrying DJ-1 mutations exhibit early onset of dyskinesia, rigidity, and tremors and respond well to L-DOPA treatment [431,434,435]. Cognitive deficits and psychotic disturbances typically arise later in disease progression [436]. The L166P mutation abolishes DJ-1 dimerization, abrogating its neuroprotective activity [437]. Post-mortem brain tissue of a patient with L172Q DJ-1 mutation exhibited Lewy body pathology [438]. The clinical manifestations of DJ-1 patients are like those with Parkin and PINK1 mutations. However, compared to Parkin and PINK1, DJ-1 mutation carriers exhibit a higher percentage of non-motor symptoms such as anxiety, cognitive decline, depression, and psychopathic symptoms [433,439,440]. It has been reported that PD brains exhibit decreased levels of DJ-1 mRNA and protein, but also show the presence of extra-oxidized DJ-1 isoforms [441] and that acidic isoforms of DJ-1 monomer were accumulated in sporadic PD brains [442]. It remains unclear how DJ-1 contributes to PD pathogenesis.

## 20. DJ-1 Induced Pluripotent Stem Cell Models

Homozygous DJ-1 mutant iPSC-derived midbrain dopaminergic neurons exhibited increased dopamine oxidation and neuromelanin-like pigmented aggregates compared to an isogenic gene-corrected control [309]. Oxidative stress from dopamine metabolism triggered mitochondrial oxidative stress, which was significantly attenuated by blocking dopamine synthesis [309]. Dopamine-induced oxidative stress inactivated GCase inhibited lysosomal function and led to increased expression of α-syn [309]. DJ-1 knockout iPSC-derived dopaminergic neurons also exhibited enhanced α-syn fibril-induced aggregation and neuronal death [443]. Other DJ-1 patient-specific iPSCs have been generated very recently, but phenotypic analysis has yet to be performed [444,445,446].

## 21. DJ-1 Animal Models

DJ-1 knockout mice show mild motor deficits and altered nigrostriatal synaptic physiology, but without dopaminergic neuron loss or significant change in dopamine levels [417,447,448]. However, one report showed that the DJ-1 knockout mouse line exhibited a loss of dopaminergic neurons in the ventral tegmental area and slight behavioral changes, such as diminished rearing behavior and impaired object recognition [449]. It was also observed that DJ-1 deficient mice exhibit alteration in dopamine metabolism, specifically an increase in dopamine reuptake causing an accumulation of striatal dopamine [450]. The higher overall level of oxidized dopamine did not result in neurodegeneration as seen in PD patients and this may be due to the fact that human SNpc dopaminergic neurons have higher overall dopamine levels than those in mice. A DJ-1 nullizygous mouse was fully backcrossed with a C57BL/6 background and displayed dramatic early onset unilateral loss of nigrostriatal dopaminergic neurons that progressed bilaterally with aging [451]. These mice also exhibited age-dependent bilateral degeneration in the locus ceruleus nucleus and displayed mild motor deficits. DJ-1 knockout mice also exhibited more microglial activation, especially in response to lipopolysaccharide insult [452]. On the other hand, DJ-1 knockout rats showed significant age-dependent nigrostriatal dopaminergic neuron loss (~50%) between 6 and 8 months of age, accompanied by motor deficits [360]. Furthermore, mitochondria from DJ-1 knockout rats showed altered respiration compared to that of WT rats [453].

## 22. Overall Discussion

As PD is primarily an idiopathic disease, the major challenge is in understanding the interplay between genetic and environmental factors that influence susceptibility to PD. In the last two decades, research into PD genetics has deepened our understanding of PD risk, onset, progression, and therapeutic approaches. These studies have revealed pathogenic pathways of neurodegeneration shared between inherited and sporadic PD. No animal PD model is perfect as it is challenging to recapitulate the age of onset, the timing of disease progression, and the spectrum of pathologies present in PD patients. Single model modalities, such as neurotoxin models recapitulate neurodegeneration and PD-like symptoms and are appropriate for addressing specific questions tailored to this model, as is the case for the genetic models. Current research has been utilizing iPSCs for a disease-in-a-dish approach. Isogenic gene-corrected cell lines offer unequivocal perturbations caused by genetic mutations. Since iPSC-based in vitro models are patient-derived they present many advantages including the ability to generate a variety of neural lineages in 3D complex systems that serve to model the in vivo brain tissue cytoarchitecture for studying pathogenesis. PD is not a single disease entity, but rather made up of subtypes based on differences in the spectrum of symptoms and the nature and distribution of Lewy body pathologies.Thus, it is necessary to combine iPSC-based in vitro models, ex vivo post-mortem brain specimens, and in vivo models to further our understanding of the different cellular and molecular mechanisms underlying PD pathogenesis.

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
