# Peer review of "Human Induced Pluripotent Stem Cell Phenotyping and Preclinical Modeling of Familial Parkinson’s Disease"

_genes, 2022, doi:10.3390/genes13111937_

Round 1

Reviewer 1 Report

Since Parkinson’s disease is the second most common neurodegenerative disease affecting more and more people and only symptomatic treatments are available, therefore the better understanding of both idiopathic and familial PD is of great importance and a huge challenge. The different models may help to reveal important aspects of the disease. The work is of great interest, a very thorough insight into the advantages/disadvantages of these models.

However, there are some comments/questions concerning the manuscript.

The authors state that “The pathological hallmark of PD is the loss of dopaminergic neurons in the substantia nigra pars compacta”. Usually the gold standard for diagnosis of Parkinson’s disease has been the presence of substantia nigra degeneration AND Lewy pathology in brain tissues (Lewy bodies consisting of alpha-synuclein) (see Kalia LV, Lang AE. Lancet. 2015, 386(9996):896-912. doi: 10.1016/S0140-6736(14)61393-3). Lewy bodies are mentioned only later (line 241). The authors also mention (line 521) that “Most PD patients with Parkin mutations do not develop Lewy body pathology [373].” In contrast to this, in lines 574-575: a parkin animal model “also lacked Lewy body pathology seen in human parkin mutant carriers”. I feel that Lewy bodies may be mentioned earlier (in the first section) and their lack in the case of parkin mutations should be discussed/clarified.  

Alpha-synuclein mutations are discussed in details, but A18T and A29S mutations should be mentioned as well. Are there data concerning iPSC for other mutations than A53T?

line 320: It is mentioned very shortly that “α-syn monomers relative to tetramers”. I think that the disordered nature of the protein, the equilibrium among monomers and tetramers, their roles and toxicity worth mentioning in a little more detail. Nowadays the oligomers are considered as the most toxic species (Ono K. Neurochem Res. 2017 Dec;42(12):3362-3371. doi: 10.1007/s11064-017-2382-x).  

lines 110 and 514-516: “The most prevalent genetic mutation has been identified in the LRRK2 gene [89,90]. It accounts for at least 1% of all PD cases and 5-15% of familial PD cases [91].” It is a little confusing. Do the authors mean that all mutations found in LRRK2 or the most common LRRK2 mutation accounts for at least 1% of all PD cases? Moreover, it is stated that “Parkin mutations account for 50% of familial and 15% sporadic cases of European PD patients with onset before 45 years of age [365,366].”

Figures/schematic representation of protein structure and the localization of the mutations may help the readers, especially in the cases of LRRK2 and alpha-synuclein.

The possible interplay between LRRK2 and GBA1 (lines 416-425) or PINK1 (lines 486-489) is very interesting. It is also mentioned (lines 447-449) that there are “several GBA1-PD models combine GBA1 mutation with overexpression of α-syn mutations to induce α-syn aggregation and pronounced PD symptoms despite the fact that these two mutations are not reported in PD patients [319,322,324].” Are there more multiple transgenic mice models available? The observations obtained from these models and the interplay between the proteins may be discussed in a separate paragraph.

line 451: Is this a separate paragraph?

Author Response

Reviewer 1

Since Parkinson’s disease is the second most common neurodegenerative disease affecting more and more people and only symptomatic treatments are available, therefore the better understanding of both idiopathic and familial PD is of great importance and a huge challenge. The different models may help to reveal important aspects of the disease. The work is of great interest, a very thorough insight into the advantages/disadvantages of these models.

However, there are some comments/questions concerning the manuscript.

The authors state that “The pathological hallmark of PD is the loss of dopaminergic neurons in the substantia nigra pars compacta”. Usually the gold standard for diagnosis of Parkinson’s disease has been the presence of substantia nigra degeneration AND Lewy pathology in brain tissues (Lewy bodies consisting of alpha-synuclein) (see Kalia LV, Lang AE. Lancet. 2015, 386(9996):896-912. doi: 10.1016/S0140-6736(14)61393-3). Lewy bodies are mentioned only later (line 241). The authors also mention (line 521) that “Most PD patients with Parkin mutations do not develop Lewy body pathology [373].” In contrast to this, in lines 574-575: a parkin animal model “also lacked Lewy body pathology seen in human parkin mutant carriers”. I feel that Lewy bodies may be mentioned earlier (in the first section) and their lack in the case of parkin mutations should be discussed/clarified.  

Thank you for these comments, although, Lewy pathology is not a requirement for diagnosis of the disease we agree that it is a gold standard and a pathological hallmark of Parkinson’s disease. We have now included this in the text.

Alpha-synuclein mutations are discussed in details, but A18T and A29S mutations should be mentioned as well. Are there data concerning iPSC for other mutations than A53T?

We have now discussed and included reference on A18T and A29S mutations. There are iPSCs for the many mutations and PD variants. However, we limited our scope to those described in the review, as we feel that the review is quite comprehensive already and won’t be able to cover everything.

line 320: It is mentioned very shortly that “α-syn monomers relative to tetramers”. I think that the disordered nature of the protein, the equilibrium among monomers and tetramers, their roles and toxicity worth mentioning in a little more detail. Nowadays the oligomers are considered as the most toxic species (Ono K. Neurochem Res. 2017 Dec;42(12):3362-3371. doi: 10.1007/s11064-017-2382-x).  

 Thank you for the comments, we have expended on this point and added reference.

lines 110 and 514-516: “The most prevalent genetic mutation has been identified in the LRRK2 gene [89,90]. It accounts for at least 1% of all PD cases and 5-15% of familial PD cases [91].” It is a little confusing. Do the authors mean that all mutations found in LRRK2 or the most common LRRK2 mutation accounts for at least 1% of all PD cases? Moreover, it is stated that “Parkin mutations account for 50% of familial and 15% sporadic cases of European PD patients with onset before 45 years of age [365,366].”

We have clarified the LRRK2 prevalence. Parkin mutations are present in early onset PD, and statistics represent the prevalence of Parkin mutations in early onset PD cases.

Figures/schematic representation of protein structure and the localization of the mutations may help the readers, especially in the cases of LRRK2 and alpha-synuclein.

Thank you for these comments, we thought about this, and we feel there is an adequate number of figures in the literature representing these genes and protein structures, and that we would’ve to do the same for all genes we describe in the review. This is something that we would address under a different topic related to drug discovery, which we are pursuing.

The possible interplay between LRRK2 and GBA1 (lines 416-425) or PINK1 (lines 486-489) is very interesting. It is also mentioned (lines 447-449) that there are “several GBA1-PD models combine GBA1 mutation with overexpression of α-syn mutations to induce α-syn aggregation and pronounced PD symptoms despite the fact that these two mutations are not reported in PD patients [319,322,324].” Are there more multiple transgenic mice models available? The observations obtained from these models and the interplay between the proteins may be discussed in a separate paragraph.

We agree that this such an exciting topic, and on its own deserve a separate review. We tried to focus the review on the principal findings and relevant preclinical models complementing the in vitro iPSC-based ones. The biology and interplay between these mutations in PD pathogenesis, is the exciting next topic.

line 451: Is this a separate paragraph?

That is correct, thank you. We have now highlighted the title in bold in a separate paragraph.

Reviewer 2 Report

The manuscript titled “Human Induced Pluripotent Stem Cell Phenotyping and Pre-2 clinical Modeling of Familial Parkinson’s disease” by Kim et al is a timely review of an important topic. I am impressed by the scope, easiness to read and comprehensiveness; however I do have some suggestions to make it better.

1.     Several important review articles need to be cited to direct readers to similar work done by others. With regard to the in vitro models, please cite the following early on:

a.      Anindita Bose, Gregory A. Petsko, and Lorenz Studer 2022 Induced pluripotent stem cells a tool for modeling Parkinson’s disease. Trends in Neurosciences 45 (8) 608-620.

b.     S. A. Antonov * and E. V. Novosadova 2021 Current State-of-the-Art and Unresolved Problems in Using Human Induced Pluripotent Stem Cell-Derived Dopamine Neurons for Parkinson’s Disease Drug Development. Int. J. Mol. Sci. 2021, 22, 3381.

2.     With regard to the in vivo genetic models, please cite the following:

a.      M. Angela Cenci and Anders Bjorklund 2020 Animal models for preclinical Parkinson’s research: An update and critical appraisal. In “Progress in Brain Research”, Volume 252, pp27-59. Elsevier B.V.

b.     Rose B. Creed, BS and Matthew S. Goldberg 2018 New Developments in Genetic Rat Models of Parkinson’s Disease. Movement Disorders, Vol. 33, No. 5, 2018

c.      Philippe Kachidian and Paolo Gubellini 2021 Genetic Models of Parkinson’s Disease. In  “Clinical Trials In Parkinson's Disease” Springer US_Humana

3.     With regard to the presentation of results, I suggest the authors follow the above-suggested review articles to tabulate what is known about the genes, mutations and possible mechanisms in the pathophysiology of PD. (The above references provide plenty of great examples.)

a.      Table 1, list of all the genes reviewed in the current manuscript;

b.     Table 2, list of genes studied the iPSC models;

c.      Table 3, list of genes studied in genetic models.

4.     The manuscript addressed the issue of what have been done very well, but lacks insight on the challenges facing researchers in the field.

Author Response

Reviewer 2:

The manuscript titled “Human Induced Pluripotent Stem Cell Phenotyping and Pre-2 clinical Modeling of Familial Parkinson’s disease” by Kim et al is a timely review of an important topic. I am impressed by the scope, easiness to read and comprehensiveness; however I do have some suggestions to make it better.

  1. Several important review articles need to be cited to direct readers to similar work done by others. With regard to the in vitro models, please cite the following early on:
  2. Anindita Bose, Gregory A. Petsko, and Lorenz Studer 2022 Induced pluripotent stem cells a tool for modeling Parkinson’s disease. Trends in Neurosciences 45 (8) 608-620.
  3. S. A. Antonov * and E. V. Novosadova 2021 Current State-of-the-Art and Unresolved Problems in Using Human Induced Pluripotent Stem Cell-Derived Dopamine Neurons for Parkinson’s Disease Drug Development. Int. J. Mol. Sci. 2021, 22, 3381.
  4. With regard to the in vivo genetic models, please cite the following:
  5. M. Angela Cenci and Anders Bjorklund 2020 Animal models for preclinical Parkinson’s research: An update and critical appraisal. In “Progress in Brain Research”, Volume 252, pp27-59. Elsevier B.V.
  6. Rose B. Creed, BS and Matthew S. Goldberg 2018 New Developments in Genetic Rat Models of Parkinson’s Disease. Movement Disorders, Vol. 33, No. 5, 2018
  7. Philippe Kachidian and Paolo Gubellini 2021 Genetic Models of Parkinson’s Disease. In  “Clinical Trials In Parkinson's Disease” Springer US_Humana

 Thank you for these suggestions, we added all these references

  1. With regard to the presentation of results, I suggest the authors follow the above-suggested review articles to tabulate what is known about the genes, mutations and possible mechanisms in the pathophysiology of PD. (The above references provide plenty of great examples.)
  2. Table 1, list of all the genes reviewed in the current manuscript;
  3. Table 2, list of genes studied the iPSC models;
  4. Table 3, list of genes studied in genetic models.

 Thank you for this great idea, we choose to do a different representation and a diverse approach based on the in vivo and on vitro models.

  1. The manuscript addressed the issue of what have been done very well, but lacks insight on the challenges facing researchers in the field.

This review addresses specific resources for experimentation. Regarding concepts and challenging aspects facing researchers that’s a different topic that would be better addressed in a different review manuscript.